# Obesogenic Diet in Mice Leads to Inflammation and Oxidative Stress in the Mother in Association with Sex-Specific Changes in Fetal Development, Inflammatory Markers and Placental Transcriptome

**DOI:** 10.3390/antiox13040411

**Published:** 2024-03-28

**Authors:** Alejandro A. Candia, Samantha C. Lean, Cindy X. W. Zhang, Daniel R. McKeating, Anna Cochrane, Edina Gulacsi, Emilio A. Herrera, Bernardo J. Krause, Amanda N. Sferruzzi-Perri

**Affiliations:** 1Centre for Trophoblast Research, Department of Physiology, Development and Neuroscience, University of Cambridge, Cambridge CB2 3EG, UK; alejandrocandiah@uchile.cl (A.A.C.); xwz20@cam.ac.uk (C.X.W.Z.); d.mckeating@griffith.edu.au (D.R.M.); ac2225@cam.ac.uk (A.C.); eg570@cam.ac.uk (E.G.); 2Institute of Health Sciences, University of O’Higgins, Rancagua 2841959, Chile; bernardo.krause@uoh.cl; 3Pathophysiology Program, Institute of Biomedical Sciences (ICBM), Faculty of Medicine, Universidad de Chile, Santiago 7500922, Chile; eaherrera@uchile.cl; 4Department for the Woman and Newborn Health Promotion, Faculty of Medicine, Universidad de Chile, Santiago 8380453, Chile

**Keywords:** obesogenic diet, pregnancy, placenta, sex, inflammation, oxidative stress

## Abstract

Background: Obesity during pregnancy is related to adverse maternal and neonatal outcomes. Factors involved in these outcomes may include increased maternal insulin resistance, inflammation, oxidative stress, and nutrient mishandling. The placenta is the primary determinant of fetal outcomes, and its function can be impacted by maternal obesity. The aim of this study on mice was to determine the effect of obesity on maternal lipid handling, inflammatory and redox state, and placental oxidative stress, inflammatory signaling, and gene expression relative to female and male fetal growth. Methods: Female mice were fed control or obesogenic high-fat/high-sugar diet (HFHS) from 9 weeks prior to, and during, pregnancy. On day 18.5 of pregnancy, maternal plasma, and liver, placenta, and fetal serum were collected to examine the immune and redox states. The placental labyrinth zone (Lz) was dissected for RNA-sequencing analysis of gene expression changes. Results: the HFHS diet induced, in the dams, hepatic steatosis, oxidative stress (reduced catalase, elevated protein oxidation) and the activation of pro-inflammatory pathways (p38-MAPK), along with imbalanced circulating cytokine concentrations (increased IL-6 and decreased IL-5 and IL-17A). HFHS fetuses were asymmetrically growth-restricted, showing sex-specific changes in circulating cytokines (GM-CSF, TNF-α, IL-6 and IFN-γ). The morphology of the placenta Lz was modified by an HFHS diet, in association with sex-specific alterations in the expression of genes and proteins implicated in oxidative stress, inflammation, and stress signaling. Placental gene expression changes were comparable to that seen in models of intrauterine inflammation and were related to a transcriptional network involving transcription factors, LYL1 and PLAG1. Conclusion: This study shows that fetal growth restriction with maternal obesity is related to elevated oxidative stress, inflammatory pathways, and sex-specific placental changes. Our data are important, given the marked consequences and the rising rates of obesity worldwide.

## 1. Introduction

The increased prevalence of pre-conceptional maternal obesity has been reviewed extensively and is established as a burden worldwide regardless of the socioeconomic status of the population [1]. This is worrying, as obesity during pregnancy poses a high risk for poor health for both mother and fetus, resulting in a higher risk of pregnancy loss, gestational diabetes, pre-eclampsia, caesarean section, instrumented vaginal birth, and preterm birth [2]. Furthermore, obesity during pregnancy elevates the risk of fetal growth restriction (FGR), large for gestational age (LGA), and metabolic alterations in the newborn [3,4].

There are several mechanisms that could be involved in the adverse maternal and neonatal outcomes associated with obesity. For instance, the physiological increase in insulin resistance during pregnancy is exacerbated in mothers with obesity, leading to pathological hyperinsulinemia and hyperglycemia in late gestation, as reported in human and non-human animal models [5,6,7,8]. Conversely, while pregnancy is characterized by wide-spread changes in the maternal inflammatory environment, the existence of obesity in the mother may lead to an imbalance in this process, as obesity is associated with a chronic low-grade inflammatory state [9,10]. Indeed, the normal circulatory increase in cytokines, such as interleukin-6 (IL-6) and tumor necrosis factor-alpha (TNF-α), during the third trimester is exacerbated in women with overweight/obesity when compared to normal-weight women [11,12]. The placenta is the primary site of communication and nutrient exchange between the mother and fetus and, hence, has been described as the primary mediator of adverse gestational conditions, like maternal obesity on fetal outcomes [13]. In experimental animals, diet-induced obesity is linked to increased placental expression of monocyte chemoattractant protein-1 (MCP-1), cluster of differentiation 14 (CD14), and 68, as well as placental macrophage infiltration [14,15,16]. Similarly, in pregnant mice with obesity, there are increased levels of inflammatory markers, including TNF-α, IL-6, interleukin-1β (IL-1β), nuclear factor kappa-light-chain-enhancer of activated B cells (NFκB) and interleukin-10 (IL-10) in the placenta [8,17,18,19]. Nonetheless, there is controversy among studies, with some reporting no changes in placental cytokine levels when comparing control and obesogenic diet-fed mice [20,21]. 

Studies have also demonstrated that obesity is related to increased levels of oxidative stress, characterized by amplified production of oxidized lipids (including oxidized low-density lipoproteins) and thiobarbituric acid reactive substances (TBARs) [22]. There are also reports of reduced levels of antioxidants, such as superoxide dismutase and glutathione, in mothers with obesity [23]. Both hyperinsulinemia and inflammation are strongly related to oxidative stress, which may also contribute to poor pregnancy outcomes in the context of maternal obesity [24,25]. In this regard, the liver is one of the main organs in the mother involved in metabolism, immune responses, and oxidative balance [26]. This organ is particularly affected by obesity and insulin resistance [27], with an increased formation of lipid droplets, a key step in the pathogenesis of non-alcoholic fatty liver disease (NAFLD) [28]. Furthermore, NAFLD is associated with altered hepatic function [29], and, in pregnancy, increases the risk of gestational diabetes, pre-eclampsia, and preterm birth [30,31]. These data suggest that the liver could be an important organ to explore in the context of poor pregnancy outcomes with maternal obesity. 

Although not fully explored, previous studies have reported that maternal obesity can prime changes in offspring health in a sex-dependent fashion [32]. For instance, in mice, pancreatic islets of female, but not male, offspring exposed to maternal obesity during intrauterine development have increased mitochondrial respiration and density [33]. Prior work has reported that the expression of metabolic genes and proteins by the placenta can also be differentially affected by maternal obesity in a sex-specific manner [34,35]. Yet, few of the studies of maternal obesity that report increased levels of oxidative stress and inflammatory markers in the placenta have explored the influence of fetal sex [8,17,19,35,36,37,38,39,40]. Likely, an imbalance in the production of inflammatory mediators and oxidative stress in obese pregnancy leads to impaired placental development and function with consequences for female and male fetal growth [41]. However, no study has precisely assessed the relationships between inflammatory pathways, oxidative stress, and placental gene expression in the context of fetal outcomes with maternal obesity.

Therefore, the aim of this study in mice was to determine the effect of diet-induced obesity on maternal hepatic lipid handling, inflammatory and redox state, as well as placental oxidative stress, inflammatory signaling and gene expression in relation to the growth of female and male fetuses. 

## 2. Materials and Methods

### 2.1. Animal Model 

Female C57Bl/6J mice at 6 weeks old were randomly assigned to a standard (Control, energy from fat, 11% and simple sugars, 7%, n = 16) or a high-fat high-sugar (HFHS, energy from fat, 38% and simple sugars, 33%, n = 17) diet. Mice had ad libitum access to food and water, and the HFHS diet was replaced every other day to maintain palatability. After 3 weeks on their respective diet, female mice were time-mated with control diet-fed males and underwent a first pregnancy and weaning to ensure fertility. At least 7 days after the end of lactation, females were then mated a second time to control diet-fed males to generate the experimental pregnancy. In total, mice were fed their respective standard or HFHS diet for ~9 weeks prior to mating. The presence of a copulation plug was considered embryonic day (E) 0.5 and mouse dams were maintained with their pre-conception diet throughout pregnancy. Maternal weight was recorded, and adiposity was assessed by time-domain nuclear magnetic resonance scanning (LF50H Minispec, Bruker, Conventry, UK) prior to mating for their experimental pregnancy and at E17.5.

### 2.2. Tissue Collection

At day E18.5, pregnant mice were anesthetized (1:1:0.16 Fentannyl [Fentadon, Dechra Veterinary Products, Northwich, UK], Midazolam [Hypnovel^®^, Roche, Welwyn Garden City, UK] and Ketamine [Ketavet, Zoetis, Leatherhead, UK] diluted in 3.26 parts water and injected i.p. with 10 µL/g) for blood collection by cardiac puncture before cervical dislocation. Maternal blood was collected into an EDTA tube. Fetuses were removed by laparotomy and decapitated. Fetuses were sexed by visual inspection and a sample of the tail was collected and used later to confirm fetal sex by PCR (Sry gene F: 5′-GTGGGTTCCTGTCCCACTGC-3′, R: 5′-GGCCATGTCAAGCGCCCCAT-3′ and autosomal PCR control gene F: 5′-TGGTTGGCATTTTATCCCTAGAAC-3′, R: 5′-GCAACATGGCAACTGGAAACA-3′). Maternal organs and individual fetuses and placentas were weighed. Blood from the male and female fetuses with the body weight closest to the average for the litter per sex was collected. Maternal and fetal blood were centrifuged at 3000 rpm to obtain plasma and serum fractions, respectively, and samples were stored at −20 °C until analysis. From a sub-set of mice (n = 6 per group), the maternal liver and one placenta per sex per litter (from the lightest female and male fetuses in the litter) were fixed in 4% paraformaldehyde (PFA) and paraffin-embedded for later sectioning and histological analysis. From a different subset of mice, the maternal liver (n = 7 per group) and micro-dissected placenta labyrinth zone (Lz, functions in nutrient transport, n = 6 per sex per group) from the female and male fetuses weighing closest to the litter average, were snap-frozen and stored at −80 °C for molecular biology and biochemical analysis (Appendix A).

### 2.3. Plasma and Serum Samples Analysis 

Maternal plasma was analyzed by the Core Biochemical Assay Laboratory (CBAL, Cambridge University Hospital NHS Foundation Trust) to determine the concentrations of aspartate aminotransferase (AST) (Siemens Healthcare, Newark, DE, USA, intra-assay % coefficient of variation [%CV] 5–2.5%), alanine transaminase (ALT) (Siemens Healthcare, Newark, DE, USA, intra-assay %CV 8.1–3.6%) and growth differentiation factor 15 (GDF15) (R&D Systems, BioTechne: Abindong, UK, intra-assay %CV 6.5–5.9%). Maternal and fetal cytokines were also assayed by CBAL using the Milliplex^®^ Mouse Cytokine Kit (#MCYTOMAG-77k, Merk Millipore, Darmstadt, Germany, intra-assay %CV < 10%). The kit enabled the quantification of IL-1α, IL-1β, IL-5, IL-6, IL-10, IL-17A, MCP-1, granulocyte/macrophage-colony stimulator factor (GM-CSF), TNF-α, IFN-γ and VEGF-A, and was performed as per manufacturers specifications and read on a MagPix^®^ plate reader (Luminex, Austin, TX, USA). Ferric Reducing Antioxidant Power (FRAP) assay of maternal plasma was done according to manufacturer instructions (FRAP Assay Kit, colorimetric, ab234626, Abcam, Waltham, MA, USA). 

### 2.4. Maternal Liver Assessments 

#### 2.4.1. Fat Content Assay

Maternal liver fat content was assessed as described previously [6]. In brief, 100 mg of homogenized tissue from the left hepatic lobule was added to 1 mL of Folch mixture (chloroform:methanol 2:1), mixed with distilled H_2_O (dH_2_O) and centrifuged at 13,200 rpm for 10 min to obtain the lipid phase. The weight of the lipid phase (lipid content) was assessed following dry extraction at 37 °C overnight and data are presented as a percentage of wet tissue mass. 

#### 2.4.2. Glycogen Content Assay

Glycogen content of the maternal liver was assessed as previously described [42]. Briefly, 100 mg of tissue in 1 mL of dH_2_O was incubated at 55 °C for 10 min with acetate buffer 0.5 M (pH 4.5). Samples were then incubated at 55 °C for a further 10 min with or without 70 U/mg amyloglucosidase enzyme. Following, 0.3 M zinc sulphate and 0.3 M barium hydroxide (Sigma-Aldrich, Saint Louis, MO, USA) were added to deproteinize the samples. After centrifugation at 3000 rpm for 10 min, glucose content of the supernatant was quantified on a YSI Analyser (YSI Inc., Yellow Springs, OH, USA). The glucose concentration of the samples in the presence and absence of amyloglucosidase were compared to quantify the conversion of glycogen to glucose as the indicator of glycogen content. Data are presented as mg/g of wet liver tissue. 

#### 2.4.3. Liver Steatosis 

The left lateral lobe of the maternal liver was sectioned at 7 µm, and hematoxylin and eosin (H and E) stained. Stained sections were randomly and systematically imaged to obtain ten representative images at 20×. Images were processed using ImageJ2 V2.14.0 (NIH, Bethesda, MD, USA) and macros that had been previously validated to assess lipid droplets size as a marker of steatosis [43]. Briefly, the original H and E image was converted to 8-bit-grey-scale image, color-inverted and a threshold of 150–255 grey levels was applied to identify the lipid droplets. Particle analysis was then configured to count particles with circularity between 0.5 and 1.0 arbitrary units (AU) and a diameter between 0.1 and 50 µm. The macros created a database with each droplet diameter and area in each image, which were compiled for the 10 images per sample from which actual and relative distributions of lipid droplet size could be assessed. 

#### 2.4.4. Liver Protein Levels 

The abundance of proteins in the maternal liver and placenta was determined as described previously [6]. In brief, 100 mg and 40 mg of frozen liver and placenta, respectively, were individually homogenized in a mix of RIPA buffer and protease inhibitors (Sigma-Aldrich, Saint Louis, MO, USA, R0278). The protein concentration was determined using a bicinchoninic acid kit (BCA, Sigma-Aldrich, Saint Louis, MO, USA, BCA1-1KT) according to the manufacturer’s protocol. The extracted protein (50 µg in 1×SDS buffer) was heated at 90 °C for 5 min and loaded into acrylamide gels before being incubated with primary antibodies against catalase (CAT, #14097, Cell signaling, Danvers, MA, USA, 1:5000), 3-nitrotyrosine (3-NT, #9691 Cell signaling, 1:10,000), 4-hydroxynonenal (4HNE, ab46545, Abcam, 1:1000), phosphorylated NFkB p65 (phospho-p65NFkB, #3033, Cell signaling, 1:1000), total NFkB, (NFkB p65, #8242, Cell signaling, 1:1000) phospho-p38-Mitogen-activated protein kinase (MAPK) (phospho-p38 MAPK (Thr180/Tyr182), #4511, Cell signaling, 1:1000), and total p38-MAPK (p38 MAPK, #8690, Cell signaling, 1:1000). Primary antibodies were detected by secondary antibody (NA934V; 1:10,000, Amersham Biosciences, Buckinghamshire, UK), developed by ECL (SuperSignal West Femto, Thermo Fisher, Waltham, MA, USA, 34095) and imaged on an iBright CL1500 imaging system (Thermo Fisher, A44114). Intensities of the bands representing the proteins of interest were determined using ImageJ2 V2.14.0 analysis software (NIH, Bethesda, MD, USA). The abundance of proteins of interest was normalized to total protein loading, as determined by quantifying of the staining of the entire lane for each sample using Ponceau-S (Thermo Scientific, A400000279). This approach, which quantified the total protein content across a broad range of molecular weights, was done to overcome potential variations in the levels of a specific band used in the normalization [44]. Detailed Western blot images are available in Appendix A. 

### 2.5. Labyrinth Zone Structure Analysis 

Placentas were histologically assessed as previously described [45]. In brief, paraffin wax-embedded placentas were sectioned at 10 μm. Sections were stained in H and E for analysis of the gross placental structure, namely, the volume density of labyrinth zone (Lz), junctional zone (Jz) and decidual basalis (Db). Other sections were stained in lectin and cytokeratin and then counter-stained with H and E to identify Lz compartments, namely fetal capillaries (FC), trophoblast (T) and maternal blood spaces (MBS), respectively. Volume densities and surface areas of these Lz components were determined, along with measurements made of the interhaemal membrane thickness.

### 2.6. Placental Labyrinth Zone RNA-Seq

Total RNA was extracted using the RNeasy Plus Mini Kit (Qiagen, Manchester, UK) according to the manufacturer’s protocol. RNA-seq libraries were prepared from 500 ng of RNA using TruSeq stranded total RNA library preparation kit (Illumina, San Diego, CA, USA) and barcoded libraries were combined and submitted for sequencing (paired-end 50 cycles on a HiSeq6000) by the Wellcome Trust-MRC institute of Metabolic Science Genomics Core Facility. Differential gene expression was performed using DESeq2. The RNA-seq data have been deposited in NCBI’s Gene Expression Omnibus and are accessible through GEO Series accession number GSE262011. Heatmaps were generated using Partek Genomics Suite V 7.20.0831. The differentially expressed genes (DEGs) were processed in The Database for Annotation, Visualization, and Integrated Discovery (DAVID analyses tool) [46,47] to identify significantly altered pathways and biological processes impacted by HFHS diet-induced maternal obesity. Gene identification can be consulted in Appendix A. 

Putative transcription factors driving the gene expression changes in the placental Lz were identified by predicted transcription factor analysis. Briefly, promoters from DEGs’ gene list were obtained from the Eukaryotic Promoter Database (EPD) [48]. Following this, the promoter list was processed using the Motif Enrichment Analysis [49] from the Motif-Based Sequence Analysis Tools (The MEME Suite 5.5.3) [50] to determine significantly enriched transcription factors. 

### 2.7. Statistics 

Data were analyzed using GraphPad Prism 9.0 (GraphPad Software, LLC, Boston, MA, USA). The ROUT test was applied to identify outliers and Shapiro–Wilk was used to determine the normality of data. When appropriate, data were analyzed by unpaired Student’s *t*-test (maternal parameters) or two-way ANOVA (using a representative or litter mean per sex for fetal and placental parameters with diet and fetal sex as variables). Data were presented as individual values or mean ± SEM, with individual sample sizes stated in the corresponding figure legend/table. Differentially expressed genes (DEG) were considered when *p*-adjusted value <0.05 and 0.5-linear fold expression change. Significant enriched transcription factors were considered when *p*-adjusted value <0.05. All other data were considered significant at *p* < 0.05.

## 3. Results

### 3.1. Maternal Body and Liver Tissue Composition, Oxidative Stress and Inflammatory Signaling

Pre-pregnancy body weight was similar (Figure 1A), but percentage adiposity was significantly higher by nearly 2-fold in females fed the HFHS diet, compared to those fed the control chow (Figure 1B). However, at the end of pregnancy (at E17.5), maternal body weight was lower in HFHS diet-fed mice, and there were no differences in adiposity between groups (Figure 1C,D). 

At E17.5, maternal liver weight, both absolute and weight adjusted after hysterectomy, was increased in HFHS dams, compared to controls (Figure 1E,F). Histological examination revealed that livers from HFHS dams showed increased steatosis markers, along with an elevated percentage fat content (Figure 1G–I), and reduced glycogen content (Figure 1J). Protein levels of catalase, an antioxidant enzyme, was reduced (Figure 1K), meanwhile, 3-NT abundance, a marker of oxidative damage to proteins, was increased in the liver of HFHS dams (*p* = 0.053; Figure 1L). In contrast, there was no change in the levels of 4HNE, a marker of lipid peroxidation in the maternal liver between groups (Figure 1M). Finally, abundance of activated phosphorylated NFκB-p65, an inflammatory-signaling protein, was similar between groups, with a tendency for increased levels of total NFκB-p65 (*p* = 0.053), and significantly decreased levels of phosphorylated to total NFκB-p65 ratio in the liver of HFHS fed mice, compared to controls (Figure 1N–P). Hepatic abundance of phosphorylated MAPK p38, a stress-signaling protein was significantly increased in HFHS mice (Figure 1Q). This increase was proportional to total MAPK p38 abundance, as phosphorylated to total MAPK p38 in the maternal liver was not significantly different between dietary groups (Figure 1R,S). These results suggest that diet-induced obesity leads to hepatic steatosis in association with increased levels of oxidative stress, inflammation, and stress signaling. 

### 3.2. Maternal Plasma Levels of Damage Markers 

The abundance of eleven cytokines in the plasma was assessed in HFHS-fed and control dams at E17.5. Of the quantified cytokines that are usually classified as pro-inflammatory, we found significantly lower concentrations of IL-17, significantly increased IL-6 (Figure 2A,B), and no change in GM-CSF, IFN-γ and TNF-α in the circulation of HFHS dams (Figure 2C–E). Regarding the anti-inflammatory cytokines analyzed, we found decreased levels of IL-5 and no change in IL-10 in HFHS compared to control dams (Figure 2F,G). Finally, the concentrations of IL-1α, VEGF-A, IL-1β and MCP-1 in maternal plasma were below the detectable levels of the assay in more than the 50% of samples in both dietary groups and were, therefore, not compared statistically (Appendix A). 

Additionally, antioxidant capacity was assessed in maternal plasma using FRAP, with similar levels in both groups of dams (Figure 2H). We also examined the circulating abundance of liver damage markers, and found increased ALT and GDF15, but no change in plasma AST levels between the HFHS and control mice (Figure 2I–K). These data suggest that HFHS-diet dams exhibited an inflammatory profile in association with liver damage. 

### 3.3. Fetal Viability and Growth

We then explored whether HFHS-induced changes in the maternal inflammatory and oxidative stress state may have implications for fetal growth. The maternal HFHS diet had no effect on the number of total conceptuses, live fetuses, resorptions, or the ratio of female-to-male fetuses (Figure 3A–D). However, fetal weight was reduced for both sexes in HFHS-fed mice (Figure 3E). Although absolute brain weight remained similar, brain weight relative to fetal weight was increased, and liver weight, as expressed in both absolute and relative terms, to fetal weight was reduced in HFHS-fed mice (Figure 3F–I). The ratio of fetal brain weight to liver weight was also increased in HFHS mice (Figure 3J). Impacts on fetal brain weight, liver weight, and the relationship between the two, were more pronounced for female fetuses compared to males. Together, these data indicate that a maternal HFHS diet induces asymmetric fetal growth restriction. Moreover, this was most severe for female fetuses in HFHS dams. 

### 3.4. Fetal Circulating Inflammatory Markers

The effect of a maternal HFHS diet on fetal circulating inflammatory cytokines was then analyzed. The analysis revealed that maternal HFHS diet reduced fetal serum GM-CSF and increased TNF-α concentrations in both males and females (Figure 3K,L). Overall, IL-6 was higher in females compared to males. However maternal diet and fetal sex interacted to determine fetal IL-6 concentrations (*p* = 0.0196), whereby serum IL-6 concentrations were higher in female versus males in control but not HFHS dams and IL-6 levels were reduced in female fetuses from HFHS dams but not males (Figure 3M). Other pro-inflammatory cytokines, namely, IL-1α and MCP-1 in the fetus, remained similar between dietary groups (Figure 3N,O). The concentration of IL-10 was similar in fetuses of both groups (Figure 3P). IFN-γ was not detectable in the circulation of fetuses from control dams but was detectable in half of the female fetuses from mice fed the HFHS diet, indicating a significant increase (Figure 3Q). The concentrations of IL-5, IL-1β, IL-17, and VEGF-A were below the detectable assay levels in most of the samples, irrespective of the dietary group; therefore, the effect of a maternal HFHS diet could not be ascertained (Appendix A). Thus, a maternal HFHS diet is associated with changes in the abundance of inflammatory markers in the fetal circulation, which, in part, is determined by fetal sex.

### 3.5. Placental Structural Development 

Changes in placental morphology in response to maternal HFHS diet were also addressed. Placental weight was significantly reduced in HFHS animals (Table 1), an effect that was significant by pairwise comparison for male fetuses (−9.5% in males and −4.7% in females). The placentas from females were overall lighter than those from males. The HFHS diet did not affect placental efficiency expressed as the fetal-to-placental ratio; however, placentas were less efficient for females compared to male fetuses overall. There was no significant effect of a maternal HFHS diet on placental Db, Lz or Jz volume. However, the volume of the Db and Lz in the placenta was significantly lower in females compared to males regardless of the maternal diet. A deeper analysis of placental Lz morphology revealed an increase in maternal blood space volume in both males (+12.4%) and females (+22.8%) from HFHS dams, although, overall, females exhibited fewer maternal blood spaces than males. The maternal HFHS diet had no impact on TB and FC volumes in the placental Lz. Lz capillary length remained similar between dietary groups, but barrier thickness was significantly reduced, by 4.5% in males and 12.2% in females, by a maternal HFHS diet. Finally, the theoretical diffusion capacity of the placenta was not altered by a maternal HFHS diet; however, this was, overall, lower in females compared to males. Collectively, these data indicate that a maternal HFHS diet impacts the morphological development of the placental Lz. 

### 3.6. Placental Oxidative Stress, Inflammation, and Stress Signaling

We assessed the abundance of oxidative stress, and inflammatory- and stress-signaling markers in the micro-dissected placental Lz of control and HFHS dams. Catalase protein levels in the Lz were not affected by a maternal HFHS diet (Figure 4A,B), whereas 4HNE, total NFkB-p65, and total MAPK p38 were increased in the placental Lz of males, but not females from HFHS dams compared to the controls (Figure 4C–H). Furthermore, the ratio of total to phosphorylated MAPK p38 was increased in the placental Lz of females from HFHS mice compared to controls, with no changes seen for male fetuses (Figure 4K,L). By comparing protein abundance between the sexes within each dietary group, we found that total MAPK p38 and phosphorylated MAPK p38 were lower in the placental Lz of females compared to males in the control, but not HFHS dams (Appendix A). Therefore, a maternal HFHS diet induces sex-specific changes in oxidative stress, inflammation, and stress signaling in the placenta Lz, whereby the male fetuses are most affected.

### 3.7. Placental Transcriptome 

To obtain further information on how the placental support of fetal development may be altered by a maternal HFHS diet, we performed RNA-seq on micro-dissected Lz samples. We performed three different DESEq2 analyses: (a) control vs. HFHS regardless of sex; (b) control females vs. HFHS females; and (c) control males vs. HFHS males. This first analysis revealed that 253 genes were differentially expressed (DEGs) by a maternal HFHS when both sexes were combined (overall). The biological processes over-represented in the DEGs altered by a maternal HFHS diet have been implicated in the immune response (15 upregulated, 28 downregulated), lipid or fatty acid metabolic process (11 upregulated, 7 downregulated), response to hypoxia and ROS (4 upregulated, 7 downregulated), ion transport (10 upregulated, 11 downregulated), nutrient transport (3 upregulated, 2 downregulated) and angiogenesis (2 upregulated, 13 downregulated) (Appendix A).

Analysis of the effect of a maternal HFHS diet for each fetal sex separately identified 95 DEGs for females and 295 DEGs for males (are shown in Figure 5A–D). This revealed only 7 genes consistently upregulated (*Prlr*, *Chil3* and *Pla2g4a*) and 30 genes consistently downregulated (*Pdgfb*, *Ager* and *Eda*) by an HFHS diet in both sexes (Appendix A). The remaining genes were specifically upregulated or downregulated in females (e.g., *Cebpb*, *Slc2a8* and *Slc9b1*) and males (e.g., *Apoe*, *Lipg*, and *Oas1b*). The DEGs downregulated in the Lz by a maternal HFHS diet in both sexes have proposed roles in osteoblast differentiation, the extracellular space, and the extracellular region (Figure 5E,F) Additional pathways were identified for the downregulated DEGs in the males; these were related to collagen biosynthesis, cell surface, and endoplasmic reticulum chaperone complex (Figure 5E,F). No pathway was significantly over-represented when analyzing the DEGs upregulated by a maternal HFHS diet in the sex-specific comparisons. 

Among all the DEGs identified for our three different analyses (overall, female, and male comparisons), 22 genes were located at X chromosome. This included genes related to the immune response (*Vegfd* & *Eda*) and ion transport (*Slc6a14* & *Gabre*), as well as those implicated in sex-specific responses of the placenta to gestational stressors (*Ogt*) (Appendix A). 

To assess how a maternal HFHS diet may modulate immune response/inflammatory pathways in the placenta, we then compared our database of DEGs to a list of DEGs in the placenta from a mouse model of intrauterine inflammation induced by gestational lipopolysaccharide (LPS) exposure [51]. This revealed that 38 (15%), 5 (5%) and 48 (16%) DEGs from the overall, female, and male comparisons with a maternal HFHS diet, respectively, overlapped with the genes differentially expressed in the placenta in response to intrauterine inflammation (Appendix A). Furthermore, we observed several DEGs related to the immune response that were altered in the same direction in both the HFHS and intrauterine inflammation datasets, including *Il1a* (upregulated), *Tgtp1* (upregulated), *Thrsp* (downregulated), and *Unc5cl* (downregulated). The expressions of *Slc10a6*, *Slc6a14* and *Rgs4*, which are implicated in ion transport, were also upregulated in both datasets (Appendix A). These results indicate that a maternal HFHS diet alters the transcriptome of the placental Lz in a fashion that depends on fetal sex and is similar to that induced by intrauterine inflammation.

### 3.8. Placental Transcription Factors Analysis

To investigate the mechanisms by which a maternal HFHS diet induces changes in the placental Lz transcriptome, bioinformatic analysis was conducted to search for the significant enrichment of binding sites for transcription factors in the promoter regions of DEGs. This revealed that transcription factors LYL1 and PLAG1 were potentially key factors driving the differential expression of genes in the overall comparison (55/253 = ~22% and 53/253 = ~21%) and that PLAG1 was a key factor for the DEGs in the male-specific comparison (46/297 = 15%) in response to a maternal HFHS diet (no significant enrichment of binding for 113 transcription factors tested at the promoters of the DEGs in females; Figure 6). The mRNA expression of genes encoding LYL1 and PLAG1 was not altered in the placental Lz by a maternal HFHS diet. These data highlight that a maternal HFHS diet may alter the placental transcriptome through modulating the activity of the transcription factors LYL1 and PLAG1. 

## 4. Discussion

The current study shows that an obesogenic HFHS diet impaired maternal lipid handling, and the inflammatory and redox state during pregnancy. This occurred along with increased fetal and placental inflammatory markers, augmented placental oxidative damage and stress signaling, changes in the placental transcriptome, and asymmetric fetal growth restriction. Furthermore, fetal, and placental changes induced by a maternal obesogenic HFHS diet were dependent on fetal sex. Females showed more pronounced changes in fetal growth and in the circulating inflammatory profile, but fewer transcriptional and stress-signaling changes in the placenta compared to the male fetuses of HFHS mice. As previous work has shown that abnormal patterns of fetal growth are associated with elevated rates of mortality and morbidity [52,53], and there are sex differences in postnatal disease susceptibility [54], these results have implications for understanding the health risks and lifespan of human populations across the globe consuming obesogenic HFHS diets. 

Maternal health was adversely affected by the obesogenic HFHS diet. Dams showed excessive lipid accumulation in the liver and markers of liver damage in their circulation (elevated ALT and GDF15), which are indicative of steatosis and NAFLD. Maternal hepatic glycogen storage was also diminished in HFHS diet-fed dams, which could have contributed to the pathogenesis of NAFLD [55] and also suggested a degree of liver cirrhosis [56]. In women, NAFLD increases the risk of developing preeclampsia and gestational diabetes [30,31,57]. Prior work has reported on impaired glucose handling in HFHS-fed pregnant dams [6,35,45], but little is known about blood pressure control. Hepatic lipid accumulation can also lead to the excessive production of free radicals and oxidative damage [58,59], which in turn leads to the activation of inflammatory signaling [60]. This is consistent with the findings of the current study that show a reduced abundance of the antioxidant enzyme CAT and a tendency for increased levels of the oxidative stress marker 3-NT (*p* = 0.053) in livers from dams fed the HFHS diet. It is also consistent with the elevated activation of p38 MAPK seen in the maternal liver of HFHS mice during pregnancy. Prior work has shown a complex relationship between p38 MAPK and liver lipid handling [61], as p38 MAPK plays roles in both the pathogenesis of, and protection against, high-fat diet-induced hepatic steatosis and NAFLD [62]. Further analyses are therefore needed to assess the contribution of p38 MAPK activation in liver lipid handling and, more broadly, the overall health of pregnant dams fed the HFHS diet. 

Pregnant dams fed the HFHS diet also exhibited reduced activation of NFκB in their livers. As NFκB is a central regulator of inflammation and cell death [63], reduced NFκB activation may reflect a compensatory or protective mechanism to minimize further disruptions to maternal liver homeostasis due to an HFHS diet. This may be particularly relevant given that dams fed the HFHS diet exhibited increased levels of proinflammatory IL-6 and decreased levels of proinflammatory IL-5 in their plasma. IL-17A, which is also pro-inflammatory, was lower in HFHS diet-fed pregnant mice. IL-17A additionally plays a role in the promotion of adipogenesis [64] and reduced IL-17A may reflect an adaptive effort to minimize further increases in adiposity in the mother fed the HFHS diet. However, other studies have also suggested that reduced IL-17A could result in negative effects on pregnancy outcomes, including compromised fetal viability [65]. In women, lower levels of IL-5 at term have also been associated with poor pregnancy outcomes, namely an increased risk of small for gestational age [66,67]. Our findings are consistent with other work that has found increased concentrations of IL-6 in pregnant mice fed an obesogenic HFD [18,21,68], and studies that repeatedly show obese women exhibit elevated circulating IL-6 during pregnancy [11,12]. IL-6 is also greater in women who develop pre-eclampsia and fetal growth restriction [69], and prior work has reported that increased levels of maternal IL-6 could alter fetal neurodevelopment [70,71]. Together, these data support the notion that diet-induced obesity impacts the maternal inflammatory state during pregnancy. However, whether the expanded adipose tissue, steatotic liver, and/or other tissues in mouse dams fed the HFHS contributed to the altered circulating inflammatory cytokine levels requires investigation. Furthermore, whether reducing the pro-inflammatory state of the mother could ameliorate the negative effects of obesity on pregnancy and offspring outcomes [72] requires study.

Although some work has reported an association between obesity and reduced fetal viability [14,73], our work showed similar litter sizes and the number of viable pups between pregnant dams fed a control and HFHS diet. However, the maternal HFHS diet induced asymmetric fetal growth restriction, which was particularly pronounced in female fetuses. In studies involving mice fed obesogenic diets during pregnancy, fetal growth outcomes are varied, with reports of increased [7,17,18], decreased [74,75,76,77], and even unchanged fetal growth [8,19,78,79]. The differences in fetal growth outcomes stem from the precise obesogenic model employed. For instance, the obesogenic diet used can contain anywhere between 30 to 60% fat content and can be fed to the mice from before, and during pregnancy, for as little as 3 and up to 14 weeks. The reduced growth of fetuses from HFHS diet-fed mice was related to increased TNF-α and decreased GM-CSF circulating levels, and to increased IFN-γ and reduced IL-6, specifically, in female fetuses. Prior work in mice has reported that GM-CSF is important for embryonic growth and nutrient uptake [80] and reduced fetal growth is seen in mice deficient in GM-CSF [81]. Findings from clinical studies have also associated changes in cord TNF-α and IFN-γ concentrations with poor fetal growth in the context of preeclampsia and pre-term birth [82,83]. There are also reports of altered umbilical cord IL-6 concentrations in pregnancy conditions. For instance, cord plasma IL-6 levels are reduced in fetal growth restriction associated with preeclampsia [84], and IL-6 is increased in the cord blood of neonates from obese mothers, but fetal growth/birthweight in these latter studies was not reported [85,86]. Recent work has also found that placental abundance of TNF-α is elevated, and IL-6 is reduced in women who are obese [87]. In control dams only, the abundance of IL-6 in the circulation of female fetuses was greater than in males. There is a deficiency of information on differences in cytokine levels between fetal sexes; therefore, the relevance of this finding is unclear. Chronic inflammation and activation of the JAK-STAT pathway due to cytokine signaling are known to modify insulin, growth hormone, and insulin-like growth factor signaling [88,89]. It would be interesting to identify the contribution of altered circulating cytokines and signaling pathways, like JAK-STAT, to the fetal growth changes observed with HFHS diet-fed pregnant mice. This is particularly relevant given that a previous investigation suggests that maternal diet-induced obesity in mice impacts the expression of key metabolic genes in the fetal liver in a manner that is partly influenced by sex [8]. Further, it would be interesting to examine whether changes in the fetal inflammatory profile with maternal obesity/obesogenic diets have implications for response/s to infections and neurocognitive outcomes in neonatal life [90,91,92], particularly as some offspring outcomes have been reported to be sex-dependent [32,34,35].

The source of cytokines in the fetal circulation is unknown. Whilst there is controversy about whether maternal cytokines can cross the placental barrier [93,94,95], changes in maternal cytokine levels were different from those observed in the fetal circulation in response to an HFHS diet. Furthermore, whilst the placenta may secrete cytokines into the fetal circulation, the expression of the genes encoding TNF-α, GM-CSF, IFN-γ and IL-6, as informed by RNA-sequencing, were not altered in the placental labyrinth by a maternal HFHS diet. However, the placenta harbors receptors for these cytokines, as well as those altered in the maternal circulation (IL-6, IL-5, IL-17A), and prior work in animal models and cultured trophoblast has demonstrated that they have the potential to modify placenta formation and function [81,96,97,98,99,100,101,102,103]. Therefore, changes in maternal and fetal cytokines could have implications for the placental phenotype observed in response to the maternal HFHS diet. 

In the current study, placentas from mice fed the HFHS diet were lighter than those fed the control diet. Despite this, there were beneficial changes in Lz morphology, with expanded maternal blood spaces and a thinner barrier for gaseous exchange, which would favor fetal growth support. Studies of other diet-induced obesity models have also observed similar adaptations in placental morphology [42,104]. Although diet-induced alterations in the placenta were similarly affected in both sexes, the placental Lz of males, but not females, exhibited elevated oxidative stress (lipid peroxidation; indicated by increased 4-HNE) in HFHS dams, which is similar to results seen in previous work [35]. Although we did not find any change in the level of CAT in the placental Lz regardless of fetal sex, work from human studies has shown that the activity of CAT, as well as other antioxidants, including superoxide dismutase, is reduced in the placenta of males, but not in the placenta of females, in obese women [105]. In the current study, the placenta of both sexes showed increased inflammatory signaling, with increased NFκB and total P38 MAPK in males and increased P38 MAPK activation in female fetuses of the HFHS diet-fed mice. Although not always analyzed according to fetal sex, prior work has also reported increased NFκB and oxidative stress in the placenta in addition to hypoxia and elevated *Tnfa* and *Il6*, suggesting a degree of placental damage due to a maternal obesogenic diet in mice [8,17]. Prior work has also reported sex-dependent differences in the expression of placental inflammatory markers, including altered macrophage activation and cytokine production in females, but not males in response to maternal obesity [18]. Further to its roles in cytokine and stress signaling, P38 MAPK is important for placental formation [106], particularly the development of the vascular spaces in the Lz. Whether the increase in P38 MAPK (total or activated) is responsible for the improved Lz development seen in both fetal sexes of obese mice is unclear. 

A maternal obesogenic HFHS diet affected the transcriptome of the placenta Lz. Although there were some genes similarly affected in the placental Lz of both sexes (with genes implicated in the extracellular region and space), some gene changes and associated pathways were specific to one fetal sex (for instance, positive regulation of osteoblast differentiation for DEGs in females and endoplasmic reticulum lumen and chaperone complex for the DEGs in males), and around three times more genes were differentially expressed in males compared to females in response to a maternal obesogenic HFHS diet (297 DEGs for males and 97 DEGs for females). These data are consistent with other work showing sex-dependent placental gene expression changes in response to an obesogenic diet in mice [107] and greater transcriptional response of the male compared to female placenta in the context of maternal obesity [8]. Among the genes differentially expressed in the placenta Lz with a maternal HFHS diet, we found genes including *Ager* and *Susd4,* which are involved in inflammation and immune-related pathways [108,109], and *Pdgfb*, which is involved in angiogenesis, similarly affected in all comparisons (downregulated regardless of sex, as well as for males and females separately). Specific knowledge of how these commonly affected genes may contribute to the placental and fetal outcomes in our model is unclear. However, *Ager* (also known as the advanced glycosylation end-product specific receptor; RAGE) is involved in trophoblast cytokine secretion [110,111], and studies of humans have shown reduced *Ager* in the placenta from women with gestational diabetes [112] but increased placental *Ager* expression in women with severe preeclampsia [113,114]. Furthermore, *Pdgfb* is a key gene involved in placental labyrinth development [115,116], and expression of *Pdgfb* was also downregulated in the placenta in a related model of maternal diet-induced obesity [117], as well as in the placenta in late gestation in mice with reduced uteroplacental perfusion pressure [118]. Thus, changes in the expression of genes including *Ager* and *Pdgfb* may be part of a common pathway through which adverse gestational environments impact placental and fetal development. 

Close inspection of the lists of genes differentially expressed with a maternal HFHS diet revealed sex-specific changes for genes involved in the immune response, ion, solute, and nutrient transport, lipid and fatty acid metabolism, oxidative stress, and angiogenesis. Of note, expression of the immune response gene *Il1a* by the placental Lz was upregulated only in the male fetuses of HFHS diet-fed dams. Loss of the main signaling receptor for IL-1α (IL1R) in the developing mouse conceptus is associated with the aberrant expression of nutrient transporter and growth-controlling genes in the placenta [119]. Studying the role of sex-dependent changes in placental *Il1a* mRNA for feto–placental development, especially given that maternal circulating IL-1α was elevated by an HFHS diet, is needed. Furthermore, the expression of more ion, solute, and nutrient transporters (*Slc* genes) were downregulated in the placental Lz of females than males, which could have implications for the more pronounced changes in fetal development with a maternal HFHS diet. For instance, the expression of *Slc26a7* is a sulfate transporter highly expressed by the Lz [120], and the disruption of placental sulfate transport leads to fetal developmental abnormalities in mice [121]. Furthermore, *Slc30a4* is needed for zinc uptake and transport and zinc deficiency leads to placental Lz malformations, including a thinner barrier for diffusion and fetal growth retardation [122].

Several of the genes that were differentially expressed in the placental Lz with the maternal HFHS diet were also altered in the placenta in a model of intrauterine inflammation [51]. This amounted to 15% of the DEGs identified in the placenta in HFHS dams regardless of fetal sex when compared to the inflammation model. Of note, several DEGs were found changed in the same direction, for example, with *Elovl4*, *Tgtp1*, and *Thrsp* being downregulated and *Il1a*, *Slc6a14*, *Slc10a6*, and *Apold1* being upregulated in the placenta in both conditions. Prior studies of the human placenta from women who are obese have also reported an alteration in the expression of genes related to immune responses, such as *IL1R2, TNFSF10*, and *MMP12* [123]. Our data are also consistent with the notion that maternal obesity and obesogenic diets create a lipotoxic placental environment in association with inflammation and the suppression of substrate transport and growth pathways [37].

Two transcription factors, LYL1 and PLAG1, were identified as potential crucial mediators of the transcriptomic changes in the placental Lz with the HFHS diet (up to 23% of DEGs in the overall or in the male-specific comparison, with no enrichment found for the female-only comparison). These data are exciting, given that PLAG1 has previously been identified as being associated with angiogenic gene expression in the placenta (e.g., *Pdgfb*, *Pparg* and *Socs3*) and is specifically increased in the placenta of male fetuses in response to gestational diabetes [124]. Other work has also found that PLAG1 is crucial in the dysregulation of genes in the human placenta linked to fetal growth restriction (e.g., *Igf2*, *Slc2a4*, and *Tcf4*) [125]. Furthermore, LYL1 has been implicated in blood vessel maturation [126] and regulates embryonic development and macrophage differentiation [127], which could have implications for the altered fetoplacental development and inflammatory profile in HFHS diet-fed mouse dams. Little is known about the factors, including maternal or placental hormones in the control LYL1 and PLAG1 expression. In humans, *PLAG1* methylation is related to leptin concentrations in cord blood [128] and *PLAG1* expression is correlated with increased body fat mass, insulin resistance, and fatty acid concentrations [129]. Further work is required to address more directly the contribution of genes differentially expressed between the placental Lz of control and HFHS diet-fed dams and the role and regulation of putative transcription factors in the fetoplacental outcomes of male and female fetuses.

The mechanisms involved in sex differences are not fully understood. Recent work in humans indicates that gene expression variation across different tissues is small between males and females, and the majority of DEGs consistent across several tissues are related to the sex chromosomes [130] and sex-biased binding of transcription factors to the regulatory elements of genes [130]. In checking through our dataset, only a few genes differentially expressed in the placenta with a maternal HFHS diet were found to be located on the X chromosome. This included *Ogt*, which was altered only in the males of HFHS mice and is in line with the dysregulation of *Ogt* expression in the placenta of males in animal models of stress [131] and in women with gestational diabetes [132]. However, there are also sex-specific epigenetic changes to autosomes that could be relevant [133] and signaling from the fetal gonads to the placenta that may contribute to this [134]. Further work is therefore required to identify the mechanisms underlying sex-specific responses of the placenta to a maternal HFHS diet observed in the current study. 

## 5. Conclusions

In summary, this study shows that a pregestational obesogenic diet causes hepatic steatosis, oxidative stress, and activation of pro-inflammatory pathways along with imbalanced circulating cytokine concentrations in the mother during pregnancy. Furthermore, fetuses were asymmetrically growth-restricted, showing sex-specific changes in circulating cytokines and alterations in the expression of genes and proteins implicated in oxidative stress, inflammation, and stress signaling (Figure 7). These data are important, given the rising rates of obesity worldwide, and the established link between exposure to adverse gestational environments/abnormal birthweight and the subsequent increased risk of the offspring to develop non-communicable diseases.

## Figures and Tables

**Figure 1 antioxidants-13-00411-f001:**
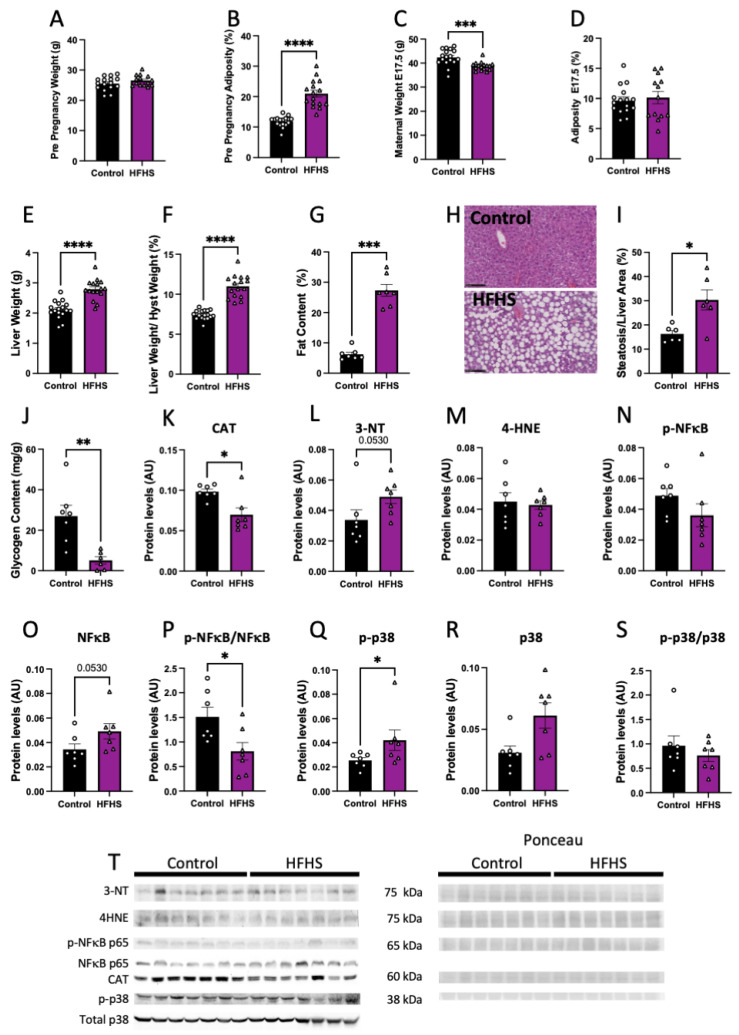
Maternal body composition and liver damage. Maternal pre-pregnancy weight (**A**); and adiposity (**B**). Maternal weight (**C**); and adiposity at E17.5 (**D**). Liver weight (**E**); liver to hysterectomy weight (**F**); percentage of fat content (**G**); representative picture of liver structure (**H**; 20× magnification; black scale bar = 100 µm); steatosis to liver area (**I**); and liver glycogen content (**J**). Maternal liver protein levels of catalase (CAT, **K**); 3-nitrotyrosine (3-NT, **L**); 4-Hydroxynonenal (4-HNE, **M**); phosphorylated p65-NFκB (**N**); total levels of p65-NFκB (**O**); phosphorylated to total p65-NFκB (**P**); phosphorylated p38 (**Q**); total p38 (**R**); phosphorylated to total p38 (**S**); and representative image of Western blots (**T**). Groups are controlled (Control, black bars, circles, n = 17 from **A**–**F**; n = 7 from **G**–**T**) and high-fat high-sugar diet (HFHS, magenta bars, triangles, n = 16 from **A**–**F**; n = 7 from **G**–**T**). Each dot represents one individual. Mean ± SEM is shown. The ROUT test was applied to identify outliers and Shapiro–Wilk was used to determine the normality of data. Data were submitted to the Student’s *t*-test or Mann–Whitney according to data distribution. * *p* ≤ 0.05; ** *p* < 0.005; *** *p* < 0.0005; **** *p* < 0.0001.

**Figure 2 antioxidants-13-00411-f002:**
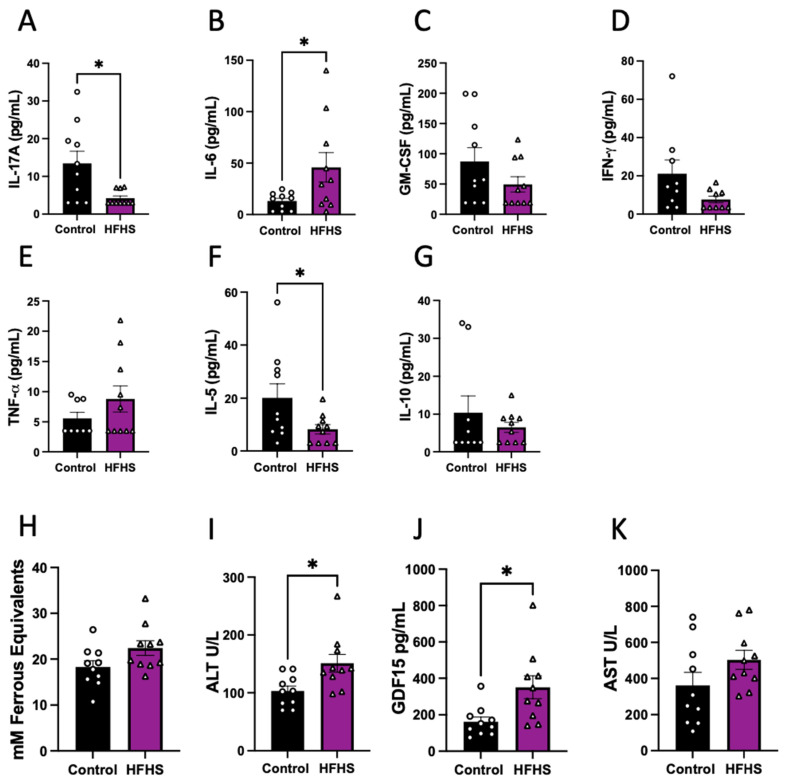
Cytokines, oxidative stress, and liver damage markers in maternal plasma. Maternal plasma levels of IL-17 (**A**); IL-6 (**B**); GM-CSF (**C**); IFN-γ (**D**); TNF-α (**E**); IL-5 (**F**); and IL-10 (**G**). Antioxidant capacity of plasma (**H**); and plasma levels of ALT (**I**); GDF15 (**J**); and AST (**K**). Groups are controlled (Control, black bars, circles n = 10) and high-fat high-sugar diet (HFHS, magenta bars, triangles, n = 10). Each dot represents one individual. Mean ± SEM is shown. The ROUT test was applied to identify outliers and Shapiro–Wilk was used to determine the normality of data. Data were submitted to the Mann–Whitney test. * *p* ≤ 0.05 vs. control.

**Figure 3 antioxidants-13-00411-f003:**
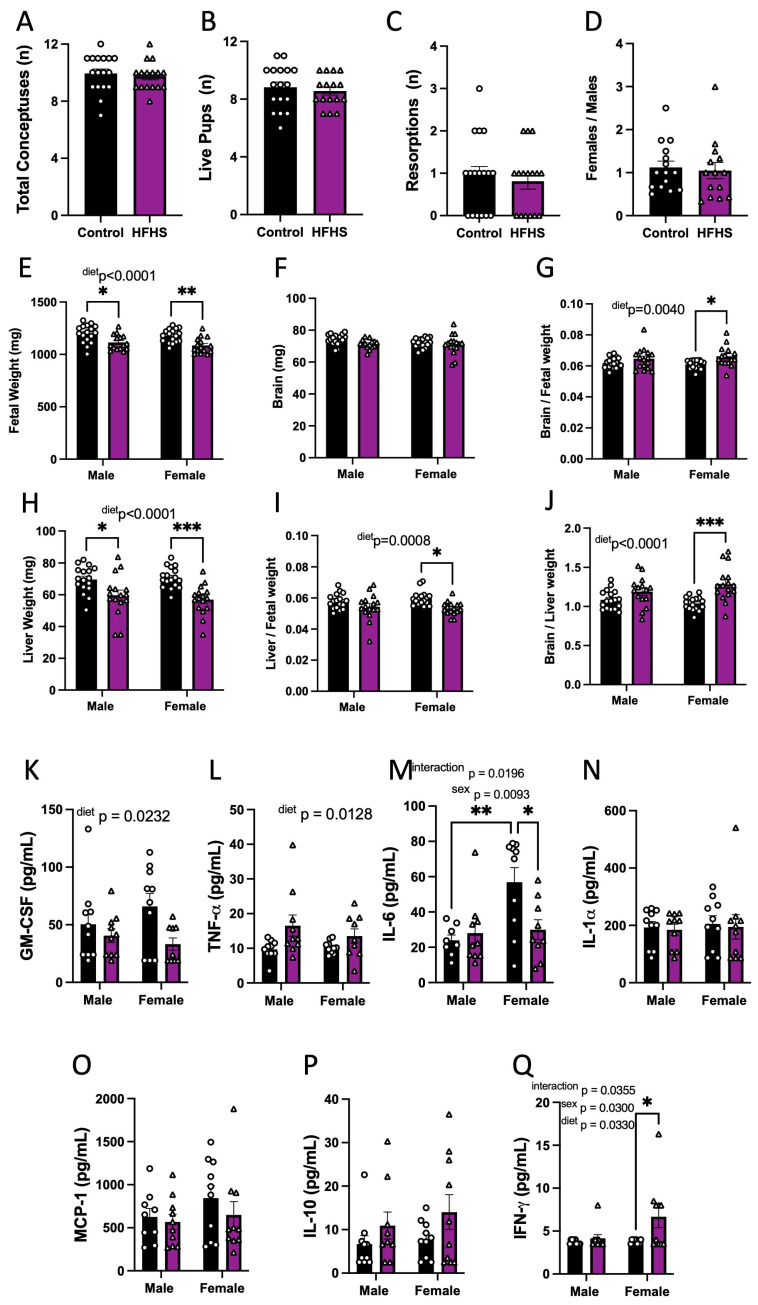
Fetal viability, growth, and inflammatory state. The total number of conceptuses (**A**); live pups (**B**); resorptions (**C**); and female-to-male ratio (**D**) at E18.5 are shown. Fetal weight (**E**); brain weight (**F**); brain-to-fetal weight ratio (**G**); liver weight (**H**); liver-to-fetal weight ratio (**I**); and brain-to-liver weight ratio (**J**) at E18.5 are shown. Fetal plasma levels of GM-CSF (**K**); TNF-α (**L**), IL-6 (**M**); IL-1α (**N**); MCP-1 (**O**); IL-10 (**P**); and IFN-γ (**Q**). Groups are controlled (Control, black bars, circles n = 17 per group from **A**–**D**; n = 17 per group/sex from **E**–**J**; n = 10 per group/sex from **K**–**Q**) and high-fat high-sugar diet (HFHS, magenta bars, triangles, n = 16 per group from **A**–**D**; n = 16 per group/sex from **E**–**J**; n = 10 per group/sex from **K**–**Q**). Each dot represents one litter (**A**–**D**; **K**), the litter mean (**E**–**J**), or individual values (**K**–**Q**). Mean ± SEM is shown. The ROUT test was applied to identify outliers and Shapiro–Wilk was used to determine the normality of data. Data were submitted to the Mann–Whitney test (**A**–**D**), or two-way ANOVA and Tukey post hoc pairwise comparison (**E**–**Q**). * *p* ≤ 0.05; ** *p* < 0.005; *** *p* < 0.0005 vs. control.

**Figure 4 antioxidants-13-00411-f004:**
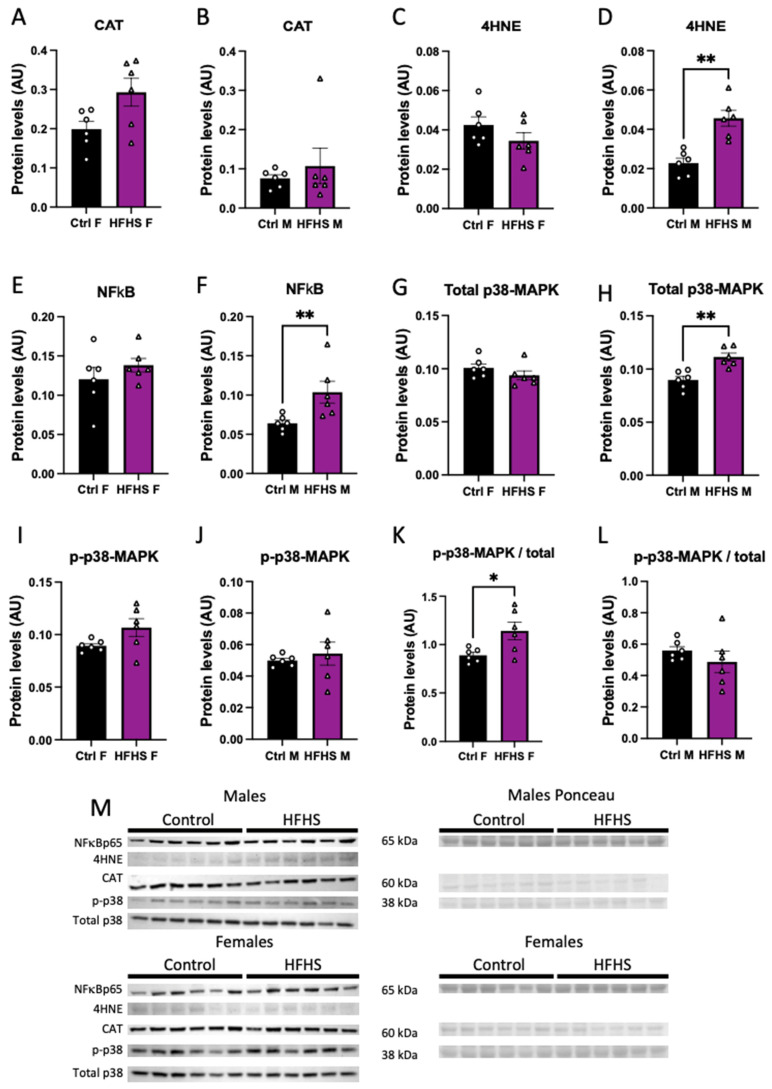
Placental labyrinth zone protein levels. Protein levels of catalase (CAT, **A**,**B**); 4-Hydroxynonenal (4-HNE, **C**,**D**); total levels of p65-NFκB (**E**,**F**); total p38 (**G**,**H**); phosphorylated p38 (**I**,**J**); phosphorylated to total p38 (**K**,**L**); and representative image of Western blots (**M**) from placental labyrinth zone of females (Ctrl F, HFHS F) and males (Ctrl M, HFHS M) are shown. Groups are control (Control, black bars, circles, n = 6) and high-fat high-sugar diet (HFHS, magenta bars, triangles, n = 6). Each dot represents one individual. Mean ± SEM is shown. The ROUT test was applied to identify outliers and Shapiro–Wilk was used to determine the normality of data. Data were submitted to the Student’s *t*-test or Mann–Whitney according to data distribution. * *p* ≤ 0.05; ** *p* < 0.005 vs. control.

**Figure 5 antioxidants-13-00411-f005:**
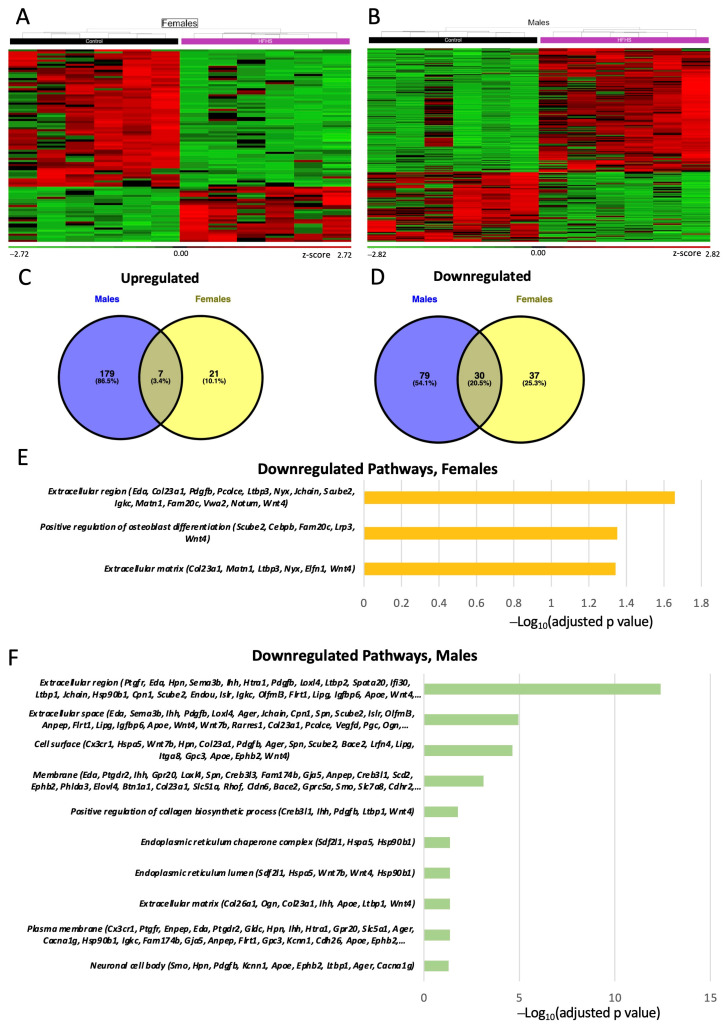
Placental labyrinth zone gene expression. Heatmap of differentially expressed genes (DEGs) in females (**A**); and males (**B**). Green and red colors indicate low and high z-score, respectively. Venn diagram of upregulated (**C**); and downregulated (**D**) differentially expressed genes (DEGs) in the placenta of female and males determined by RNA-seq (*p* adjusted value < 0.05 and 0.5-linear fold expression change). Pathways significatively downregulated (*p* adjusted value < 0.05) by HFHS diet in the placenta of females (**E**); and male fetuses (**F**) determined by FDR < 0.5. Data are n = 6 per group per sex.

**Figure 6 antioxidants-13-00411-f006:**
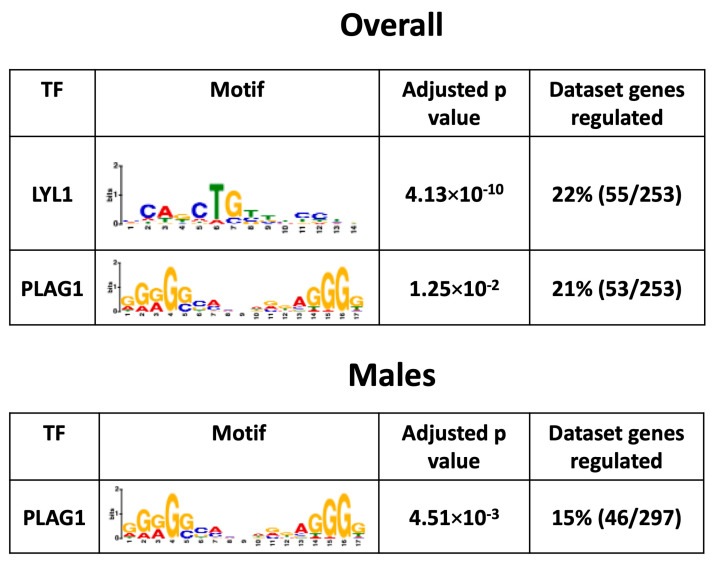
Predicted enriched transcription factors. Significantly enriched transcription factors, their motif, adjusted *p*-value, and percentage of genes regulated in each dataset from each comparison are shown.

**Figure 7 antioxidants-13-00411-f007:**
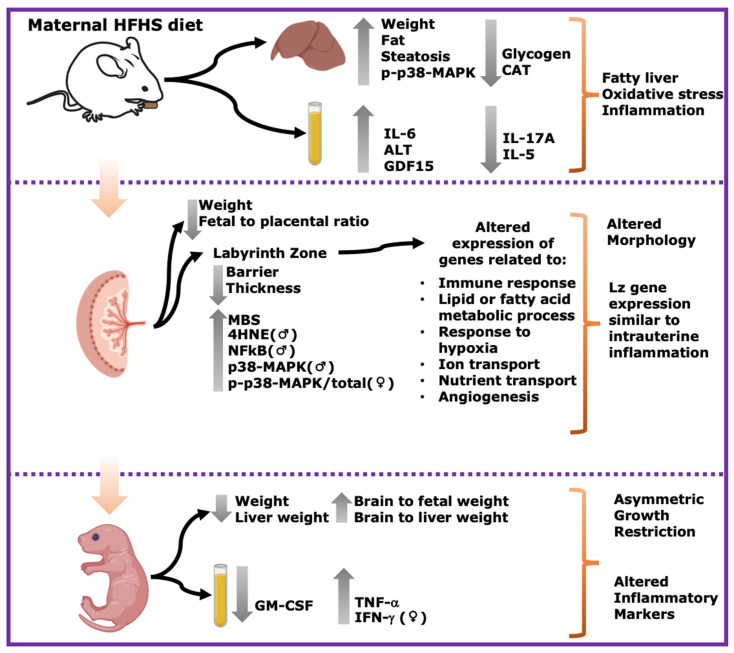
Summary of main results. Maternal HFHS diet induces fatty liver, oxidative stress and inflammation in dams; alters morphology and Lz gene expression; and induces asymmetric growth restriction and altered inflammatory markers in fetuses. Grey arrows pointing upwards indicate significant increase in HFHS group. Grey arrows pointing downwards indicate significant decrease in HFHS diet group. ♀: significant changes only in females. ♂: significant changes only in males.

**Table 1 antioxidants-13-00411-t001:** Placental biometry and structure. Db: decidua basalis; Jz junctional zone; Lz: labyrinth zone; MBS: maternal blood space; TB: trophoblast; FC: fetal capillaries; TDC: theoretical diffusion capacity. Mean ± SEM is shown. The ROUT test was applied to identify outliers and Shapiro–Wilk was used to determine the normality of data. Data were submitted to two-way ANOVA and Tukey post hoc pairwise comparison. Bold numbers represent significant differences (*p* ≤ 0.05 vs. control).

	Control						HFHS						*p* Value		
	Male			Female			Male			Female					
Structure or Measurement	Mean	±	SEM	Mean	±	SEM	Mean	±	SEM	Mean	±	SEM	Diet	Sex	Interaction
Placental Weight (g)	0.10	±	0.0023	0.09	±	0.0012	0.09	±	0.0026	0.09	±	0.0018	**0.0018**	**<0.0001**	0.2049
Fetal to Placental Ratio	11.91	±	0.3451	12.81	±	0.3555	11.82	±	0.3263	12.40	±	0.3394	0.2474	**0.0089**	0.3822
Db, mm^3^	17.09	±	1.3630	12.65	±	1.2330	15.62	±	1.5460	11.79	±	0.9515	0.3895	**0.0051**	0.8213
Jz, mm^3^	32.87	±	2.2400	29.86	±	4.1160	30.61	±	2.7510	32.92	±	2.1070	0.8902	0.9035	0.3634
Lz, mm^3^	46.34	±	2.2340	39.56	±	1.2790	44.24	±	1.0410	40.83	±	2.1900	0.8160	**0.0084**	0.3496
Lz—FC, mm^3^	19.93	±	1.7370	16.38	±	0.9625	16.65	±	0.9153	17.03	±	1.5020	0.3403	0.2512	0.1578
Lz—MBS, mm^3^	11.30	±	0.5420	8.53	±	0.2379	12.70	±	0.8470	10.48	±	0.7709	**0.0190**	**0.0010**	0.6786
Lz—TB, mm^3^	15.11	±	0.7809	14.65	±	0.9512	14.90	±	0.7750	13.31	±	1.1350	0.4019	0.2718	0.5404
Capillary Length, µm	67.62	±	2.0190	60.81	±	1.6510	65.70	±	4.6930	63.97	±	6.1490	0.8794	0.3014	0.5353
Barrier Thickness, µm	3.24	±	0.0905	3.63	±	0.1157	3.10	±	0.1389	3.18	±	0.1912	**0.0424**	0.1014	0.2900
TDC, cm^2^*min^−1^*kPA^−1^	0.03	±	0.0021	0.01	±	0.0011	0.03	±	0.0026	0.01	±	0.0006	0.9389	**<0.0001**	0.8935
Surface Area, cm^2^	0.05	±	0.0022	0.02	±	0.0008	0.04	±	0.0023	0.02	±	0.0011	0.4452	**<0.0001**	0.3730
Capillary Diameter, µm	19.23	±	0.6491	18.48	±	0.3735	18.11	±	0.7099	18.49	±	0.7150	0.3969	0.7729	0.3905

## Data Availability

Data are contained within the article. RNA-seq raw data files are accessible through GEO Series accession number GSE262011.

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
