# Peer review of "Obesogenic Diet in Mice Leads to Inflammation and Oxidative Stress in the Mother in Association with Sex-Specific Changes in Fetal Development, Inflammatory Markers and Placental Transcriptome"

_antioxidants, 2024, doi:10.3390/antiox13040411_

Round 1

Reviewer 1 Report

The manuscript Candia et al. “Obesogenic diet in mice leads to inflammation and oxidative stress in the mother in association with sex-specific changes in fetal development, inflammatory markers and placental transcriptome” describes the effect of obesogenic diet in mice on parameters of insulin resistance, inflammation and oxidative stress. The authors studied changes in maternal liver and plasma, placenta and in fetal serum using broad spectrum of methods including RNA-seq, measurement of liver parameters and histological analysis of placenta. The results are well-structured and clearly presented. Provided graphic material is also clear. The results contribute to better understanding of negative consequences of obesogenic diet on fetus. 

I only have several minor comments.

Can you fill in concentration of Ponceau S used?

Figure 1B and D shows comparison of adiposity. How did you measure it? Can you mention it in the Methods?

Figure 1B – typo in the description of axis Y (adipocity)

Page 7, line 274 – please add “increased” between “showed” and “steatosis”

Page 11, 3.4 – can you explain what do you mean by interaction? (line 345)

Table 1 description (line 383) – “placental structure” is redundant.

Page 18 – different fonts are used in the second paragraph.

Author Response

Reviewer 1.

Major comments

The manuscript Candia et al. “Obesogenic diet in mice leads to inflammation and oxidative stress in the mother in association with sex-specific changes in fetal development, inflammatory markers and placental transcriptome” describes the effect of obesogenic diet in mice on parameters of insulin resistance, inflammation and oxidative stress. The authors studied changes in maternal liver and plasma, placenta and in fetal serum using broad spectrum of methods including RNA-seq, measurement of liver parameters and histological analysis of placenta. The results are well-structured and clearly presented. Provided graphic material is also clear. The results contribute to better understanding of negative consequences of obesogenic diet on fetus. 

Answer: Thank the reviewer for their kind comments.

Detail comments

I only have several minor comments.

  1. Can you fill in concentration of Ponceau S used?

Answer: We used Ponceau S Staining Solution (Thermo Scientific; A40000279) without dilution according to manufacturer’s instructions. The company do not report the concentration of the solution.

  1. Figure 1B and D shows comparison of adiposity. How did you measure it? Can you mention it in the Methods?

Answer: It was measured by time-domain nuclear magnetic resonance (TD-NMR). This detail has been added (see line 117)

  1. Figure 1B – typo in the description of axis Y (adipocity)

Answer: Thank you for this observation. We have fixed Figure 1B.

  1. Page 7, line 274 – please add “increased” between “showed” and “steatosis”

Answer: Thank you for this comment. We have added “increased” (line 282)

  1. Page 11, 3.4 – can you explain what do you mean by interaction? (line 345)

Answer: By interaction we are referring to the result of our 2-way ANOVA which allows us to identify the effect of each variable and their interaction on our outcome measures. The nature of the interaction can result in opposite or synergistic changes to a variable when the two variables are analysed together compared to when only one variable is analysed.

In section 3.4, there was a significant interaction of diet and fetal sex on IL-6 concentrations (p=0.0196). In this case, IL-6 was greater for females compared to males in control dams, but not in HFHS dams. Also, the HFHS diet only reduced IL-6 in the females but not the males.  We have amended the sentence to make it clearer.

  1. Table 1 description (line 383) – “placental structure” is redundant.

Answer: Thank you for this observation. We have deleted “placental structure” (line 393)

  1. Page 18 – different fonts are used in the second paragraph.

Answer: Thank you for this comment. We have changed fonts in that paragraph (lines 518-541)

Reviewer 2 Report

This study investigated the effects of maternal high-fat diet on fetal growth and placental development. The results showed that maternal high-fat diet led to fetal growth restriction, altered placental structure and function, and sex-specific effects on the placenta. The study provides evidence that maternal high-fat diet has adverse effects on fetal growth and placental development, which may be related to health problems in offspring after birth. The study employed a variety of research techniques to investigate nucleic acids and protein biomarkers in maternal serum, fetal serum, and placental tissue. The results are solid, with a large amount of experimental data. However, there are still some issues that need to be clarified and the manuscript need to be revised before publication.

Figures 1 and 4:

 Although Ponceau S staining can be used to quantify proteins, it is not ideal for use as a loading control in Western blotting. Ponceau S staining produces background signal and may not bind completely to proteins, leading to inaccurate quantification. It is possible for two samples to have the same amout of proteins at the cut strips, but have different amounts of total proteins. In contrast, a housekeeping protein is a protein that is expressed at a constant level in all cells, making it a more reliable indicator of loading. Thus, Ponceau S staining of bands at the same molecular weight height on a Western blot may not be an ideal loading control as compared to using a housekeeping protein.

Results 3-3, 3-5

They found that female placentas were more susceptible to the adverse effects of maternal high-fat diet than male placentas. One possible explanation for the sex-specific difference in placental response to maternal high-fat diet is that there are sex-specific differences in placental gene expression. Trophoblasts are derived from the embryo, and X chromosome expression is known to be sexually dimorphic. However, the authors did not mention whether they have identified some genes located in sex chromosomes or indirectly regulated by genes located on sex chromosomes.

Figure 6 and Discussion:

They have also identified two transcription factors, LYL1 and PLAGL1, as potential drivers of the gene expression changes in response to maternal HFHS diet. However, the authors have not further analyzed whether these two transcription factors regulate genes on the sex chromosome, which could potentially influence trophoblast and placental development. Additionally, they have not discussed whether maternal or placental hormones and growth factors could affect gene expression through these two transcription factors.

Figure 5A, 5B:

The Venn diagram in Figure 5 is illogical. The authors used three circles to represent the relationship between three sets of samples: overall samples (blue), male samples (green), and female samples (yellow). However, the use of three circles is unnecessary and illogical. For example, in the data for genes upregulated in Figure 5A, there is a gene that is upregulated in both female and male samples, but does not show a differential expression in the overall samples. A similar situation occurs in the data for downregulated genes in Figure 5B. This logical fallacy is due to the authors' misuse of the Venn diagram in the analysis of gender differences. A two-circle Venn diagram would be sufficient to represent the relationship between the three sets of samples. If a two-circle Venn diagram were used, then the intersection between male and female samples should be considered gender-independent, possibly stemming from maternal regulation. Meanwhile, the non-intersecting areas of male and female samples could be explained by gender influence. Viewing it from this perspective would avoid the paradox mentioned above, and it is recommended that the authors make this correction.

Figure 5C, 5D:

The authors used DAVID to perform enrichment analysis, but only presented the p-values (EASE Score) and not the FDR (false discovery rate) values. FDR is a more important metric for assessing the significance of enrichment results, as it accounts for the multiple hypothesis testing problem and can be used to estimate the probability of false positives.

Figure 5:

Additionally, the authors did not report the number of replicates they performed for each analysis, which is important information for assessing the reliability of the results. It is better to present a heatmap of gene expression, which would allow readers to visualize the overall pattern across different samples.

This study investigated the effects of maternal high-fat diet on fetal growth and placental development. The results showed that maternal high-fat diet led to fetal growth restriction, altered placental structure and function, and sex-specific effects on the placenta. The study provides evidence that maternal high-fat diet has adverse effects on fetal growth and placental development, which may be related to health problems in offspring after birth. The study employed a variety of research techniques to investigate nucleic acids and protein biomarkers in maternal serum, fetal serum, and placental tissue. The results are solid, with a large amount of experimental data. However, there are still some issues that need to be clarified and the manuscript need to be revised before publication.

Figures 1 and 4:

 Although Ponceau S staining can be used to quantify proteins, it is not ideal for use as a loading control in Western blotting. Ponceau S staining produces background signal and may not bind completely to proteins, leading to inaccurate quantification. It is possible for two samples to have the same amout of proteins at the cut strips, but have different amounts of total proteins. In contrast, a housekeeping protein is a protein that is expressed at a constant level in all cells, making it a more reliable indicator of loading. Thus, Ponceau S staining of bands at the same molecular weight height on a Western blot may not be an ideal loading control as compared to using a housekeeping protein.

Results 3-3, 3-5

They found that female placentas were more susceptible to the adverse effects of maternal high-fat diet than male placentas. One possible explanation for the sex-specific difference in placental response to maternal high-fat diet is that there are sex-specific differences in placental gene expression. Trophoblasts are derived from the embryo, and X chromosome expression is known to be sexually dimorphic. However, the authors did not mention whether they have identified some genes located in sex chromosomes or indirectly regulated by genes located on sex chromosomes.

Figure 6 and Discussion:

They have also identified two transcription factors, LYL1 and PLAGL1, as potential drivers of the gene expression changes in response to maternal HFHS diet. However, the authors have not further analyzed whether these two transcription factors regulate genes on the sex chromosome, which could potentially influence trophoblast and placental development. Additionally, they have not discussed whether maternal or placental hormones and growth factors could affect gene expression through these two transcription factors.

Figure 5A, 5B:

The Venn diagram in Figure 5 is illogical. The authors used three circles to represent the relationship between three sets of samples: overall samples (blue), male samples (green), and female samples (yellow). However, the use of three circles is unnecessary and illogical. For example, in the data for genes upregulated in Figure 5A, there is a gene that is upregulated in both female and male samples, but does not show a differential expression in the overall samples. A similar situation occurs in the data for downregulated genes in Figure 5B. This logical fallacy is due to the authors' misuse of the Venn diagram in the analysis of gender differences. A two-circle Venn diagram would be sufficient to represent the relationship between the three sets of samples. If a two-circle Venn diagram were used, then the intersection between male and female samples should be considered gender-independent, possibly stemming from maternal regulation. Meanwhile, the non-intersecting areas of male and female samples could be explained by gender influence. Viewing it from this perspective would avoid the paradox mentioned above, and it is recommended that the authors make this correction.

Figure 5C, 5D:

The authors used DAVID to perform enrichment analysis, but only presented the p-values (EASE Score) and not the FDR (false discovery rate) values. FDR is a more important metric for assessing the significance of enrichment results, as it accounts for the multiple hypothesis testing problem and can be used to estimate the probability of false positives.

Figure 5:

Additionally, the authors did not report the number of replicates they performed for each analysis, which is important information for assessing the reliability of the results. It is better to present a heatmap of gene expression, which would allow readers to visualize the overall pattern across different samples.

Author Response

Reviewer 2.

This study investigated the effects of maternal high-fat diet on fetal growth and placental development. The results showed that maternal high-fat diet led to fetal growth restriction, altered placental structure and function, and sex-specific effects on the placenta. The study provides evidence that maternal high-fat diet has adverse effects on fetal growth and placental development, which may be related to health problems in offspring after birth. The study employed a variety of research techniques to investigate nucleic acids and protein biomarkers in maternal serum, fetal serum, and placental tissue. The results are solid, with a large amount of experimental data. However, there are still some issues that need to be clarified and the manuscript need to be revised before publication.

Answer: Thank you for your comments. We have tried our best to address and clarify the comments of the reviewer.

  1. Figures 1 and 4:

Although Ponceau S staining can be used to quantify proteins, it is not ideal for use as a loading control in Western blotting. Ponceau S staining produces background signal and may not bind completely to proteins, leading to inaccurate quantification. It is possible for two samples to have the same amout of proteins at the cut strips, but have different amounts of total proteins. In contrast, a housekeeping protein is a protein that is expressed at a constant level in all cells, making it a more reliable indicator of loading. Thus, Ponceau S staining of bands at the same molecular weight height on a Western blot may not be an ideal loading control as compared to using a housekeeping protein.

Answer: Thank you for your comment. We understand your concern regarding the use of Ponceau, however, its employment has been validated previously (PMID: 20206115) and is widely used in scientific papers. We specifically chose to use Ponceau as we have found that house-keeping proteins can be responsive to dietary manipulations. In addition, an advantage of the Ponceau is that you can assess total protein loading, by analysing the entire lane, rather than just a protein of a certain molecular weight which may not be representative of the protein of interest. We have added a few more details about the analysis of the Ponceau in line 209.

  1. Results 3-3, 3-5:

They found that female placentas were more susceptible to the adverse effects of maternal high-fat diet than male placentas. One possible explanation for the sex-specific difference in placental response to maternal high-fat diet is that there are sex-specific differences in placental gene expression. Trophoblasts are derived from the embryo, and X chromosome expression is known to be sexually dimorphic. However, the authors did not mention whether they have identified some genes located in sex chromosomes or indirectly regulated by genes located on sex chromosomes.

Answer: Thank you for your observation. We agree that genes expressed on the sex chromosomes could contribute to the sexually dimorphic responses seen with diet.

In checking through our dataset, only a few DEGs identified are located on the X chromosome (we have included them in Supplementary table 4). This includes Ogt which was altered just in the males of HFHS mice and is line with the dysregulation of Ogt expression in the placenta of males in animal models of stress (PMID: 24979775) and in women with gestational diabetes (PMID: 36442303). However, there are also sex-specific epigenetic changes to autosomes that could be relevant (PMID: 26241064) and signalling from the fetal gonads to the placenta that may contribute (PMID: 28523122). We have added a paragraph at the end of the discussion too (Lines 440-444 & 692 -704).

  1. Figure 6 and Discussion:

They have also identified two transcription factors, LYL1 and PLAGL1, as potential drivers of the gene expression changes in response to maternal HFHS diet. However, the authors have not further analyzed whether these two transcription factors regulate genes on the sex chromosome, which could potentially influence trophoblast and placental development. Additionally, they have not discussed whether maternal or placental hormones and growth factors could affect gene expression through these two transcription factors.

Answer: Thank you for your comment. We have checked using JASPAR Predicted Transcription Factor Targets and other databases, and none of the DEGs located on the X chromosome are predicted targets of LYL1 and PLAG1. Based on our literature searches, little is known about the regulation of LYL1 and PLAG1 expression by maternal and placental hormones. Prior work has shown that Plag1 methylation is related to leptin concentration in cord blood (PMID:31851703) and Plag1 gene expression is correlated with increased body fat mass, insulin resistance and fatty acid concentrations (PMID:34063412). We have included a comment about these studies and the need for further work in the revised discussion (see line 686 – 687).

  1. Figure 5A, 5B:

The Venn diagram in Figure 5 is illogical. The authors used three circles to represent the relationship between three sets of samples: overall samples (blue), male samples (green), and female samples (yellow). However, the use of three circles is unnecessary and illogical. For example, in the data for genes upregulated in Figure 5A, there is a gene that is upregulated in both female and male samples, but does not show a differential expression in the overall samples. A similar situation occurs in the data for downregulated genes in Figure 5B. This logical fallacy is due to the authors' misuse of the Venn diagram in the analysis of gender differences. A two-circle Venn diagram would be sufficient to represent the relationship between the three sets of samples. If a two-circle Venn diagram were used, then the intersection between male and female samples should be considered gender-independent, possibly stemming from maternal regulation. Meanwhile, the non-intersecting areas of male and female samples could be explained by gender influence. Viewing it from this perspective would avoid the paradox mentioned above, and it is recommended that the authors make this correction.

Answer: Thank you for your comment. We apologise for any confusion. We have now provided a better description of how our RNAseq analysis were performed. In particular, we performed three different DESEq2 analyses: a) Control vs HFHS regardless of sex (Control [n=12] vs HFHS [n=12]); b) Control females vs HFHS females (Control females [n=6] vs HFHS females[n=6]); and c) Control males vs HFHS males (Control males [n=6] vs HFHS males [n=6]). In checking through our RNAseq datasets, we found that the two genes that were seen in the sex-specific comparisons but not the overall comparisons (Pnoc and Gm13269) had been removed from the overall comparison during the initial filtering step with the raw data. We do not have an explanation for why it was removed from the overall but not the sex-specific comparisons, especially as the filtering steps were exactly the same for analyses. We believe it may be to do with the different sample sizes between the overall and the sex specific comparisons. Due to this oversight, and to be consistent between our datasets, we have removed these two genes from the sex specific comparisons. However, we agree that the three-circle Venn diagram may be confusing and as requested, we only show the two-circle Venn diagram representing the intersection between male and female samples for the HFHS diet effect. We have also adapted the results text to first describe the results of the overall and then the sex-specific comparisons. See lines 419-420 & 429-434. We also report the sample size for the RNAseq analyses in relevant tables and figures of the revised paper.

  1. Figure 5C, 5D:

The authors used DAVID to perform enrichment analysis, but only presented the p-values (EASE Score) and not the FDR (false discovery rate) values. FDR is a more important metric for assessing the significance of enrichment results, as it accounts for the multiple hypothesis testing problem and can be used to estimate the probability of false positives.

Answer: Thank you for this observation. Previously we had selected the genes based on FDR but we plotted the -Log10(p adjusted value) as we felt that it aided data visualization. However, as recommended, we have modified the charts according to your suggestion and show -Log10FDR (Figure 5 C & D).

  1. Figure 5:

Additionally, the authors did not report the number of replicates they performed for each analysis, which is important information for assessing the reliability of the results. It is better to present a heatmap of gene expression, which would allow readers to visualize the overall pattern across different samples.

Answer: Thank you for your comment. We have included the sample number for each comparison in figure legend (Lines 446-450). We have also included heat maps of the DEGs per sex. See revised Figure 5

Round 2

Reviewer 2 Report

The authors have addressed concerns related to Ponceau S staining, sex-specific gene expression, transcription factor analysis, Venn diagrams, enrichment analysis, and heatmap of gene expression. They have implemented appropriate revisions to enhance the clarity and quality of the manuscript. Nonetheless, minor aspects remain that necessitate further refinement prior to publication.

Figures 1 and 4:

In the cited paper (PMID: 20206115), quantification was performed across a broad range of molecular weights instead of focusing on a specific band. The practice of using a local band for normalization introduces an issue: the selected band might be affected by variations in the target protein, compromising its utility as a control for total protein. To align with the practices outlined in the referenced study and to address these concerns, it is recommended to discuss the limitations associated with using a local band for normalization and to underscore the significance of evaluating the entire lane. Additionally, inclusion of the full-lane Ponceau S stain image in the main text, or guidance directing readers to its location in the supplementary files, is advised. This would facilitate a comprehensive understanding of total protein levels across samples.

Figure 5:

In the previous version of the manuscript, the authors labeled plots as '-Log10(p-value)', instead of '-Log10(adjusted p-value)'. In the revised version, labeling as '-Log10FDR' is incorrect, as FDR (False Discovery Rate) and adjusted p-value represent distinct concepts; FDR denotes the ratio of false positives among all significant findings, whereas adjusted p-value refers to a p-value corrected for multiple testing. It is advised to label these as '-Log10(adjusted p-value)' to accurately represent the authors' findings.

lines 229-230 and Data Availability Statement:

Regarding the lines 229-230, the statement, 'The RNA-seq data have been deposited in NCBI's Gene Expression Omnibus and are accessible through GEO Series accession number GSEXXXXXX (will be uploaded after paper approval)', is improper at this stage of publication. The authors should provide the actual accession number in the manuscript. Furthermore, the Data Availability Statement on line 744, 'Data is contained within the article', is incomplete and misleading. Given that the RNA-seq data will be deposited in GEO, this statement should be revised to reflect the GEO Series accession number.

Author Response

Major comments

The authors have addressed concerns related to Ponceau S staining, sex-specific gene expression, transcription factor analysis, Venn diagrams, enrichment analysis, and heatmap of gene expression. They have implemented appropriate revisions to enhance the clarity and quality of the manuscript. Nonetheless, minor aspects remain that necessitate further refinement prior to publication.

Detail comments

Figures 1 and 4:

In the cited paper (PMID: 20206115), quantification was performed across a broad range of molecular weights instead of focusing on a specific band. The practice of using a local band for normalization introduces an issue: the selected band might be affected by variations in the target protein, compromising its utility as a control for total protein. To align with the practices outlined in the referenced study and to address these concerns, it is recommended to discuss the limitations associated with using a local band for normalization and to underscore the significance of evaluating the entire lane. Additionally, inclusion of the full-lane Ponceau S stain image in the main text, or guidance directing readers to its location in the supplementary files, is advised. This would facilitate a comprehensive understanding of total protein levels across samples.

Answer: We have modified the text accordingly (line 211-214). Also, we have provided a supplementary file where we showed the western blots in detail (line 735-736) and we indicated it in the text (line 214).  

Figure 5:

In the previous version of the manuscript, the authors labeled plots as '-Log10(p-value)', instead of '-Log10(adjusted p-value)'. In the revised version, labeling as '-Log10FDR' is incorrect, as FDR (False Discovery Rate) and adjusted p-value represent distinct concepts; FDR denotes the ratio of false positives among all significant findings, whereas adjusted p-value refers to a p-value corrected for multiple testing. It is advised to label these as '-Log10(adjusted p-value)' to accurately represent the authors' findings.

Answer: We modified the graph including the -Log10(adjusted p-value) as requested. Please see figure 5.

lines 229-230 and Data Availability Statement:

Regarding the lines 229-230, the statement, 'The RNA-seq data have been deposited in NCBI's Gene Expression Omnibus and are accessible through GEO Series accession number GSEXXXXXX (will be uploaded after paper approval)', is improper at this stage of publication. The authors should provide the actual accession number in the manuscript. Furthermore, the Data Availability Statement on line 744, 'Data is contained within the article', is incomplete and misleading. Given that the RNA-seq data will be deposited in GEO, this statement should be revised to reflect the GEO Series accession number.

Answer: We have provided the GEO accession number in line 234. Also, we have clarified it in line 747. We will change the embargo date after the paper is published or when the editor suggests us to do it. In case the reviewers want to check the data please follow the instructions below:

To review GEO accession GSE262011:
Go to https://www.ncbi.nlm.nih.gov/geo/query/acc.cgi?acc=GSE262011
Enter token mxyxsmgcphavpad into the box

Round 3

Reviewer 2 Report

The Ponceau S staining revisions, statistical label correction, and GEO accession number inclusion greatly improve manuscript quality and data transparency. Please release the GEO dataset publicly upon publication.

The Ponceau S staining revisions, statistical label correction, and GEO accession number inclusion greatly improve manuscript quality and data transparency. Please release the GEO dataset publicly upon publication.